# Runway: A novel, forward-screen method of analyzing catwalk gait analysis data

**Dana Creasman**[1,2]*, **Aileen Anderson**[1,2,3]

**1** Sue and Bill Gross Stem Cell Research Center, Irvine, United States of America, **2** Department of Anatomy and Neurobiology, Irvine, United States of America, **3** Institute for Memory Impairments and Neurological Disorders, Irvine, United States of America

* dcreasma@uci.edu

## Abstract

Assessing locomotor endpoints is a crucial component of neurotrauma research. One method of measuring locomotor behavior is through gait analysis, where one empirically measures different aspects of animal locomotion. A common tool for performing this analysis is via using the CatWalk kinematic testing apparatus, which utilizes data captured from freely moving animals traversing a tunnel with a transparent walkway, specially lit and filmed from below to create a high-contrast video of paw placement during locomotion. The CatWalk software outputs high dimensional data, which presents an issue when attempting to determine what the sum of the data means in terms of locomotor outcomes. To address this concern, we have developed *runway,* an R package for the processing, analysis, visualization, and ultimately interpretation of Catwalk data. The *runway* pipeline uses a novel, forward-screen method for analysis, systematically identifying variables affected by the treatment or intervention in question and analyzes the isolated variables through various methods to pinpoint how animal gait is affected. We explain in detail how *runway* processes experimental data, and provide a pair of use cases to demonstrate *runway's* power as an analytical tool.

## Introduction

Assessing locomotor function is a critical aspect of spinal cord injury (SCI), traumatic brain injury (TBI), and other neurotrauma and neurodegeneration research. Over the history of neurotrauma research, numerous scales and tasks have been developed, each with their respective strengths, weaknesses, and intended uses. Live-graded, observation-based tasks such as BBB and BMS are sensitive, but ordinal scaling of these tasks results in an uneven, non-linear progression of recovery through the scale [1,2]. For example, several points on the BBB and BMS scales are dedicated to changes in limb movement, while a difference of only one point separates an animal that can step vs. one that cannot [1,2]. This creates regions of scale compression and

**Data availability statement:** All relevant data are within the paper and its Supporting Information files.

**Funding:** This research was funding by NIH grants #1R01NS123927-01A1, #NS082174, and #5R01NS117103.

**Competing interests:** The authors have declared that no competing interests exist.

scale expansion, which can warp the analysis and interpretation of data [3]. Other tasks such as the Horizontal ladderbeam (or similar tasks like the pegged beam or wire grid) operate on a linear scale, scoring either successful steps or errors made during walking [4–10]. This task is highly sensitive, but only over a narrow region of recovery, as animals must be consistently stepping enough to perform the task, but not stepping so well as to create a ceiling effect [4]. Additionally, while the clear "better/worse" dichotomy present in tasks such as BBB, BMS, and ladderbeam has the benefit simplifying interpretation of performance, this simplicity is also a downside, as it does not allow for robust analysis of how performance is achieved [1,2,4–6,11].

These issues highlight the relevance and importance of gait analysis. The goal of gait analysis is to empirically measure different aspects of animal locomotion, such as stride length, speed, stepping pattern, etc. Early methods of obtaining this information consisted of dipping the paws of an animals in ink and letting them run across a sheet of paper, then physically measuring the desired variables from the tracks [12,13]. Currently, advanced machine-learning techniques are being developed that allow raw video of animal locomotion to be turned into kinematic information [14–16]. Sitting in between these two ends of the technological scale is the CatWalk kinematic testing apparatus. The CatWalk suite utilizes data captured from freely moving animals traversing a tunnel with a transparent walkway, specially lit and filmed from below to create a high-contrast video of paw placement during locomotion. Videos are semi-automatically analyzed by a software package (with human quality control), providing detailed, robust insight into a myriad of aspects of how an animal is moving [17].

Like most gait analysis methods, the number of variables that CatWalk outputs is both a strength and a drawback: when dozens of variables are present, determining what the sum of the data means can be a difficult and confusing task. The potential for cherry-picking variables which show the "desired" outcome is high, as is the chance of missing potentially insightful results simply by never looking at the data in total, and instead focusing only on a few "gold standard" variables. Some approaches have gone further, attempting to use the massive data output from CatWalk to generate a linear model, correlated to another, established measure of locomotor function, such as BMS/BBB or ladder beam [18,19]. While useful under certain circumstances, this approach has two primary issues. One is that animal locomotion is highly dependent on factors such as sex, age, strain, genetics, and injury/disease model [20–22]. This means that whatever model is generated may only work under a narrow set of conditions. A second is that this approach ultimately seeks to reduce Catwalk data to a binary axis of "better" and "worse", usually with the hope of showing that a given intervention was efficacious in promoting recovery post-injury. This misses an opportunity to use Catwalk data to answer a more nuanced question: do different injuries and/or treatment alter an animal's locomotion in different ways? Post-injury recovery is multifaceted; animals can exhibit functional recovery in ways that diverge from uninjured performance. Catwalk data allows examination of how various interventions result in different compensatory mechanisms, offering insight into treatment mechanisms of action. Notably, while our examples analyses in this paper are limited to

catwalk datasets, the benefits our method are applicable to other methods of obtaining locomotor tracking data, such as deep learning techniques like DeepLabCut [14,15,23]. Our approach is primarily concerned with the interpretation of data, and thus can be coupled to any method that outputs multiple features describing animal locomotion.

To address these issues, we have developed an R package named *runway* for the processing, analysis, visualization, and ultimately interpretation of Catwalk data. The *runway* pipeline uses a novel, forward-screen method for analysis, enabling a more comprehensive and unbiased examination of animal locomotion following traumatic injury experiments. This method systematically identifies variables affected by the treatment or intervention in question and analyzes the isolated variables through various methods to pinpoint how animal gait is affected. To demonstrate the method's flexibility, we apply it to two unique use cases: 1) Analysis of a data set with animals from two groups (CD44 WT and KO, measured at 3 time points (pre-injury, 14 and 28 weeks post-injury); 2) A single-time point SCI dataset with three binary variables and eight unique groups.

## Methods

### Data processing and statistics

Catwalk datasets analyzed in the use cases was obtained using CatWalk XT (Noldus v9.0). Full info on the studies these datasets were sourced from can be found in their originating articles [24,25]. Analysis was performed using R version 4.0.3 within R Studio. ANOVA modeling was performed using the *aov* function in base R. Additional packages used in either runway or elsewhere in the R scripts were the *A86, tidyverse, scales, ggVenn, umap, ggwordcloud,* and *patchwork* packages [26–33]. Complete R scripts and accompanying data for each use case are available in the supplementary files.

### Data compilation, cleaning, and processing

The first step in the analysis is compiling all relevant CatWalk data. Ideally, data will have been obtained at multiple time points, at least before and after injury. To facilitate data import and ensure proper blinding, it is best to run each animal using only its numerical identifier, without entering any group information into CatWalk. In parallel, maintain a master lookup table that matches each ID to the animal group and other important metadata.

Once CatWalk data has been collected and master lookup table created, this data can be imported into the *runway* R package, which will handle the remaining steps automatically. *Runway* will annotate each set of CatWalk data with group info by aligning it to the lookup table, combine the individual time points into a continuous data set, eliminate any unusable variables (such as variables with all missing or all 0 values), and normalize the data. The standard way to normalize the data is by using a baseline time point, typically the pre-injury time point. However, in some cases, it might be more appropriate to choose a time point such as post-injury but pre-intervention instead. If an individual subject lacks data at the normalization time point, *runway* provides two options: either completely exclude the subject from the analysis if the study power is sufficiently high or use the group means for each variable in place of the individual value. Lastly, *runway* automatically excludes the normalization time point from subsequent analyses to prevent distortion of results.

### Isolation of significant variables using analysis of variance models

Once data is imported and pre-processed, *runway* will handle the task of determining the gait variables affected by the study's independent variables. The user inputs their desired model, typically involving the interaction of a grouping variable (e.g., treatment, genotype) and the time point post-injury. *Runway* then applies the chosen model to each normalized CatWalk variable in the complete dataset. The p-values from this analysis help identify gait variables affected by single independent variables or their interactions.

Choosing the appropriate alpha value to measure significance is important. In analyses such as these involving evaluating several variables, it is typically best practice to adjust the alpha value accordingly to compensate for the multiple

comparisons and prevent false positives. However, in this case, there is frequently an issue of low power vs. high dimensionality. Indeed, the dimensionality is often inflated by the fact that CatWalk provides many variables which are highly related to one another or otherwise superfluous. Given the exploratory and descriptive goals of this analysis method, we consider not adjusting for multiple comparisons and using an alpha value appropriate for single variable comparisons as acceptable.

## Results

As described under Methods, *runway* produces a set of affected gait variables for each independent variable (and variable interaction) in the study. Here we address an overview of these results, as well as several case examples for application and interpretation of the data generated, as well as how this process may provide value added for data interpretation.

There are several methods for interpreting these results that can be used in parallel with each other. One method is to simply evaluate the number of significant variables extracted for each independent variable. In the case of a combinatorial treatment study (such as the one covered in Use Case 2), if one treatment variable affects multiple gait variables and another does not, this could be considered a proxy measure for the efficacy of one treatment method versus another. Similarly, if each treatment variable extracts multiple gaits variables, but the interaction of the two variables does not, this implies that the mechanisms alter locomotion via parallel, non-synergistic pathways.

For both this reason, and to gain a more thorough understanding of the effects of each independent variable, it is necessary to examine the nature of the singular gait variables extracted, beginning by graphing each variable to see how they are affected by time, treatment, genotype, etc. Runway can also automatically fragment each variable into "tags" (e.g., "right forepaw" or "couplings") and perform a hypergeometric test to determine if certain tags are significantly affected. This will reveal how exactly gait has been affected. Perhaps animals in a given group are disproportionately shifting their weight onto their fore or hind paws, exhibiting a particular stepping pattern, or moving more steadily. If a given treatment affects an intuitively related set of variables, such as contralateral paw variables or variables relating to limb speed, this can help elucidate the mechanism via which an intervention is altering animal gait, informing what molecular targets or neurological pathways to target in subsequent studies.

While we argue catwalk analysis is best suited for descriptive, annotative approaches, it's often desirable to determine which group of animals is walking "best," particularly when more traditional locomotor tasks, such as BMS or ladder beam, fail to discriminate between groups. The main challenge with using Catwalk in this manner is defining what "good" walking looks like, which can be deceptively difficult.

One method involves examining a restricted set of relatively well-understood, easy-to-interpret variables. For example, healthy animals tend to spend a lot of their gait time with their paws in a "diagonal" orientation, with two paws off the ground at any given time. In contrast, injured animals often struggle with balance and thus spend more time on three or four paws. This restricted approach can be used alongside the more open and exploratory one described earlier to obtain both a coarse and fine picture of animal recovery. However, it is crucial not to assume that a given variable is inherently indicative of recovery without first validating this fact for the strain/injury/time point post-injury being analyzed, ideally by correlating it with one of the more straightforward locomotor tasks.

Another approach is to examine which group most closely resembles their pre-injury baseline, as intuition suggests this should correlate with better outcomes. However, this is a dangerous assumption, as post-injury recovery involves a high degree of compensation. Thus, a treatment that helps animals to properly compensate may, in fact, cause them to look less like their pre-injury baseline. However, baseline performance must still be taken into consideration, especially when dealing with comparisons across strain, age, time post-injury, and other factors where gait differences can be expected even at baseline (particularly with parameters like step length and step duration). This is why the initial step of normalization is so critical, along with checking the raw, non-normalized data for differences at baseline.

A more nuanced version of comparing animals to baseline is to compare their response to treatment against the directional trend seen in historical data from prior studies where recovery was observed. This is especially pertinent if a similar mechanism of recovery is expected from one intervention to the next, for example, trying an identical treatment at different doses/time points. UMAP, a dimensionality reduction tool, is effective for comparing multiple groups, including historical data from studies with observed recovery.

A graphical generalization of this entire approach can be found in Fig 1. Next, we will examine a pair of use cases for the *runway* package to provide examples of how this analysis method can provide novel insights from Catwalk data in an efficient manner.

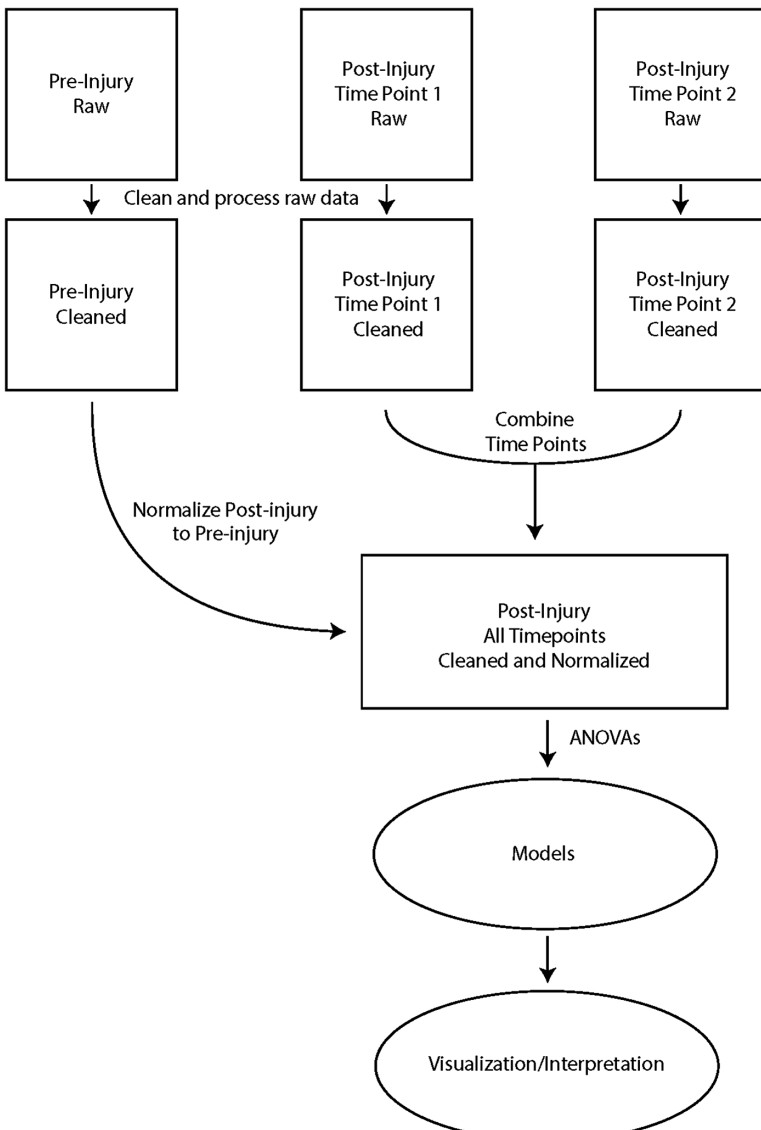

**Fig 1. Graphical summary of analysis method utilized by runway package.** Raw CatWalk data from separate time points is processed, then post-injury data is normalized to pre-injury data and combined. Normalized post-injury data is then analyzed using the appropriate ANOVA model, and then the results are visualized and interpreted.

**Case one: A multiple time point data set with one independent variable**

In the first example, the experimental paradigm involved mice of two different genotypes (WT and CD44 KO) that were given unilateral C5 contusion injuries. CatWalk data was obtained pre-injury, and at 14 and 28 days post-injury (dpi). First, data was normalized to the pre-injury time point. It should be highlighted that this normalization to baseline, while a standard part of analysis of any catwalk dataset, is particularly important in cases like this where animals are from two different genotypes and thus can be expected to possess baseline differences from one another.

After this normalization, a repeated measures ANOVA was performed to compare the effects of time vs. genotype on animal gait. A p-value of 0.05 was considered significant. This resulted in 31 variables significantly affected by time, another 31 affected by genotype, and only 1 affected by the interaction of time and genotype. A heatmap and/or a Venn diagram (Fig 2 A-B) were used to quickly visualize this information. It was concluded that while the progression of time after an injury and animal genotype both significantly impacted locomotor outcomes, these variables did so independently of one another. In other words, the two genotypes had different locomotor outcomes, but genotype did not affect how locomotion changed from 14 to 28 dpi. This implies that the differences between genotypes were already apparent by 14 days, indicating CD44 KO altered something about the early phase of the SCI response.

Next, the analysis focused on examining the affected variables, specifically regarding the effects of genotype. Three of the affected variables were "Diagonal Paw Support Percentage", "Three Paw Support Percentage", and "Lateral Paw Support Percentage", all measures of which paw animals had in contact with the ground at any point. This implies a difference in genotypes in terms of how they distributed their weight. Related to this, several variables measuring the behavior of the right-fore and left-hind paws were observed. This asymmetry aligns with the fact that the injury model was a unilateral cervical injury primarily impeding right forepaw function. Given that the left hind paw and right fore-paw contact the ground in a coupled manner during healthy animal movement, it makes sense that impacting the right forepaw would also affect the left hind paw. Similarly, several "couplings" variables were extracted, which measure how

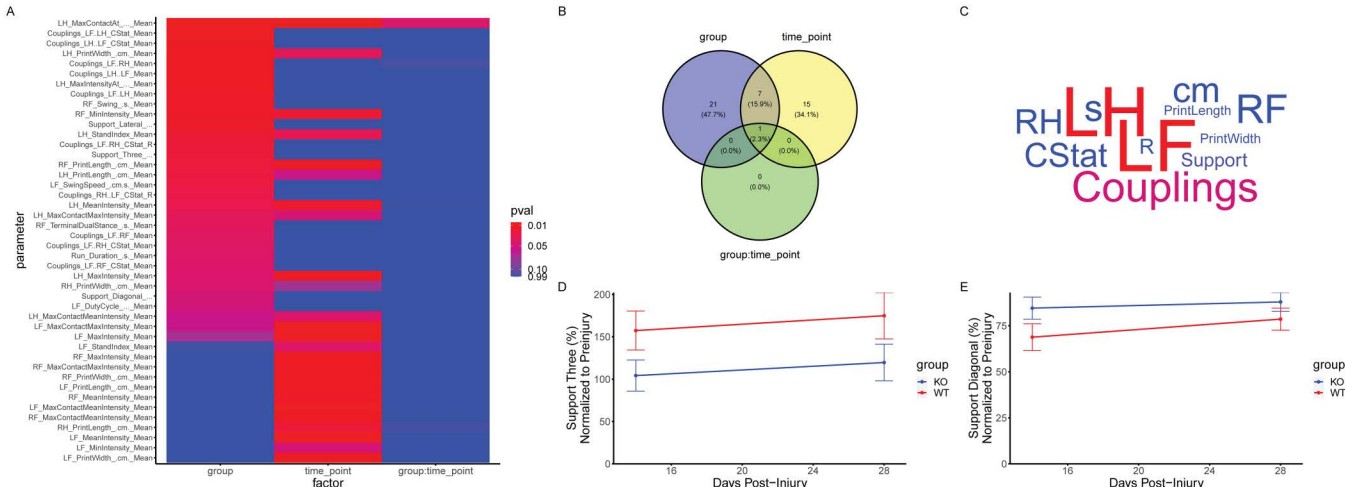

**Fig 2. Summary of the findings from analysis of case one. A)** Heatmap of isolated significant variables from repeated measures ANOVA of group vs. time, indicating that a high number of variables were affected by both time and group (columns 1 and 2, respectively), but few to no variables exhibit a combinational effect where time group alters how a variable changes over time. **B)** Venn diagram showing the number of "hits" (variables with a p-value less than 0.05) from each of the three columns in the heatmap in **A. C)** Word cloud visualization of the most common strings from the "hit" variables in the "group" category. Redder coloring indicates disproportionate hits vs. expecting as determined by a hypergeometric test, indicating that the group primarily affected left paw parameters. **D-E)** Line graphs showing the progression through time of two parameters identified by this analysis, "support percent three" and "support percent diagonal", showing that animals in the knockout group were spending more time in a diagonal arrangement than WT counterparts.

movement is coordinated between paws. Critically, however, the tags identified via a hypergeometric test as being most affected by genotype was actually "left forepaws", despite ladder beam data indicating that left forepaw function was not impaired by injury. This suggests the effect of CD44 KO primarily altered how animals compensated post-injury with their unimpaired limbs.

Looking closely at how these variables were affected by CD44 KO revealed that CD44 KO had an overall beneficial effect on locomotion. CD44 KO animals spent more time with a "diagonal" paw contact arrangement and less time with a "three" arrangement compared to WT mice, indicating a better ability to balance and support their weight. This aligned with results from a ladder beam task, which showed fewer errors committed by CD44 KO mice vs. WT mice.

Taken together, these data show that CD44 KO had a beneficial effect on animal locomotion post-SCI and suggest that this benefit was likely due to neuroprotective modulation of the early response to SCI. Indeed, histology revealed the recruitment of immune cells was reduced in CD44 KO mice, along with reductions in lesion and astroglial scar volume. This study provides an excellent example of the power of this method of gait analysis. Using CatWalk data in combination with the Runway package, we were able to go beyond asking a simple yes or no question on animal recovery and instead provide insightful detail on exactly when and how the genetic manipulation was affecting the response to SCI.

### Case Two: A single-time point SCI dataset with 3 independent variables

The second case uses data from another SCI study, where animals were given a unilateral hemisection injury, followed by treatment with four unique combinations of two treatments: bio-material bridge implementation and hNSC transplantation. In this case, CatWalk data was only obtained at a terminal time point, so pre-injury/pre-transplantation normalization was not possible. As in case one, ANOVAs were used to isolate relevant variables; hence, much of the methodological approach used was identical to that described in case one. However, in this case, a single measure, two-way ANOVA was employed for each of the independent variables. Next, the Catwalk variables affected by each individual variable were isolated (Fig 3 A-B). Critically, treating the data as having three independent variables rather than four unique groups allowed us to determine the unique ways in which each treatment affected gait. Interestingly, the results showed that cell treatment altered five variables, most dealing with right paws, whereas bridge affects ten completely different variables, mostly dealing with left paws. This suggests that the two treatments affected animal gait via different, parallel mechanisms, a fact that was independently verified via other histological and behavioral data from the study.

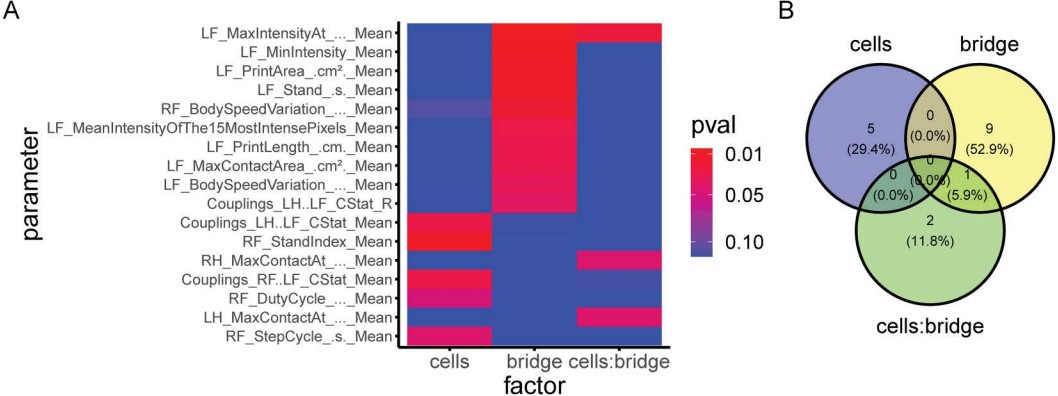

**Fig 3. Summary of the findings from analysis of case two. A)** Heatmap of isolated significant variables from repeated measures ANOVA of group vs. time, indicating that several of variables were affected by both cells and bridge (columns 1 and 2, respectively), relatively fewer variables exhibit a combinational effect. **B)** Venn diagram showing the number of "hits" (variables with a p-value less than 0.05) from each of the three columns of the heatmap, further illustrating that cells that cells and bridge treatment affected isolated sets of 4 and 6 variables, respectively, with no overlap.

It's worth noting that one variable exhibited a significant combinatorial effect, where treatment with cells altered how the variable was affected by bridge treatment. This might seem in contradiction to the earlier statement about each treatment affecting a completely distinct subset of variables, though it is not.

In this context, main effects are the effects that either bridge treatment or cell transplantation alone have on a given CatWalk parameter, ignoring the impact of the other invention. An interaction effect in this case would examine how the presence of bridge treatment impacts the effect cell transplantation has on a given CatWalk parameter, and vice versa. Interaction effects do not require the presence of main effects. That is to say, an independent variable can significantly impact the effect another independent variable has on a dependent variable, even though the overall impact both independent variables have on the dependent variables is insignificant. In terms of how to interpret this outcome, for this case, given that only one variable showed interaction vs several variables showing main effects for only one independent variable, the simplest conclusion is that the two treatments were working via separate mechanisms. This method of analysis provided critical insights into each intervention's mechanism of effect. We could not only determine what treatments or combinations were efficacious but also if the effects observed were due to overlapping/synergistic mechanisms or parallel/additive ones. The Runway R package greatly facilitates this process, making designing and applying the right model to a given dataset quite straightforward and flexible. Additionally, Runway allows for additional visualizations (such as heatmaps, word clouds, etc.) that grant greater insight into a dataset than a simple list of p-values would. When this work is published, the package will be made publicly available via GitHub in parallel.

## Discussion

Gaining a clear picture of subject locomotion is crucial in research involving locomotor impairment. CatWalk is a robust tool for obtaining a wealth of behavioral data from animal studies, but analyzing this vast data can be challenging, with the potential for incorrect conclusions or overlooked findings. We present a novel schema for analyzing CatWalk data, using an unbiased, forward-screening approach to emphasize the unique impact of injuries, diseases, or interventions on locomotor behavior, and introduce the Runway package to facilitate this analysis.

While this method provides valuable insights into animal locomotion, it has drawbacks. It is primarily descriptive of animal behavior and less robust in answering whether an intervention has improved or worsened recovery. We recommend coupling CatWalk analysis with other behavior tasks, like ladderbeam or BMS, to examine both coarse and fine aspects of behavior.

Beyond the technical statistical methods, this paper advocates for a more nuanced approach to interpreting data from CatWalk and similar gait analysis programs. The specific approach may not be perfectly suited to every experiment or tool, but the overall philosophy of exploratory analysis and asking refined questions about behavior can be applied across any study, maximizing insights from the data obtained. Critically, this approach is not dependent on using catwalk data as the input. Catwalk data can be supplemented by other factors such as score on other behavioral tasks, age, and weight gain/loss over time to examine how these factors reflect or influence recovery. Data from alternative automated and semi-automated gait analysis systems could also be used, such as from the aforementioned DeepLabCut [14,15]. Catwalk data is an excellent example use case for this method, but it is not the only option. The overall goal of this approach and the accompanying tools is to provide a means of taking gait data and generating insights to aid in interpreting a study. Even as the technical methods of collecting gait data evolve and become more sensitive, the need for data interpretation will remain, and this method will continue to be relevant.

Long-term goals of this work include updating and strengthening the *runway* package, making it more powerful and accessible for individuals in the neurotrauma field and related disciplines. Another goal is to promote public sharing of animal behavioral data and standardization, akin to genomics and transcriptomics fields. Projects like the Open Data Commons for Spinal Cord Injury share this data-sharing ethos and have begun to make more and more data from SCI experiments publicly available. Using tools like *runway* to perform meta-analyses and data mining on this data will provide

critical new insights, allowing researchers to see the broad, over-arching trends present in the field's cumulative data. By promoting such crosstalk and providing tools for in-depth analysis, we hope to see a surge in knowledge gained and treatments developed.

## Transparency, rigor, and reproducibility statement

Raw data and R scripts from this study will be made available in a FAIR data repository (odc-sci.org) prior to publication under a creative commons attribution license (CC-BY 4.0). The Runway package for R can be accessed at https://github.com/dcreasma/runway.

## Supporting information

**S1 Data. Raw Data and R Scripts.**
(ZIP)

## Acknowledgments

The authors would also like to thank Usha Nekanti for providing the data for use case 2.

## Author contributions

**Conceptualization:** Dana Creasman, Aileen Anderson.

**Data curation:** Dana Creasman.

**Formal analysis:** Dana Creasman.

**Funding acquisition:** Aileen Anderson.

**Investigation:** Dana Creasman.

**Methodology:** Dana Creasman.

**Project administration:** Aileen Anderson.

**Resources:** Dana Creasman, Aileen Anderson.

**Software:** Dana Creasman.

**Supervision:** Aileen Anderson.

**Validation:** Dana Creasman.

**Visualization:** Dana Creasman.

**Writing – original draft:** Dana Creasman.

**Writing – review & editing:** Dana Creasman, Aileen Anderson.

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
