## [Decision Letter · Decision Letter 0]

10 Mar 2025

Dear Dr. Anderson,

Thank you for submitting your manuscript to PLOS ONE. After careful consideration, we feel that it has merit but does not fully meet PLOS ONE’s publication criteria as it currently stands. Therefore, we invite you to submit a revised version of the manuscript that addresses all of the points raised during the review process.

Reviewer 1 and myself have thoroughly looked at the manuscript and we think it provides a new opportunity for CatWalk users and also for those who are deeply interested in the analysis of altered animal locomotion patterns.   

We look forward to receiving your revised manuscript.

Kind regards,

Antal Nógrádi, M.D., Ph.D., D.Sc.

Academic Editor

PLOS ONE

3. In the online submission form, you indicated that [Data will be made available upon acceptance.].

Additional Editor Comments (if provided):

Reviewers' comments:

Reviewer's Responses to Questions

**Comments to the Author**

1. Is the manuscript technically sound, and do the data support the conclusions?

Reviewer #1: Yes

2. Has the statistical analysis been performed appropriately and rigorously?

Reviewer #1: Yes

3. Have the authors made all data underlying the findings in their manuscript fully available?

Reviewer #1: Yes

4. Is the manuscript presented in an intelligible fashion and written in standard English?

Reviewer #1: Yes

Reviewer #1: Kinematic analysis is an important decent method to monitor regeneration and the efficacy of therapeutic efforts following injuries or diseases affecting the motor system. The authors of the manuscript aimed to use a new R package to make CatWalk data processing more reliable. The manuscript is technically sound, it presents clear logic and a proper scientific English.

I would like to make a few comments to improve the manuscript.

Major points

-The authors should support their statements with references in the Introduction much more often (e.g.: line 37, 44, 64). To find the place of this method among the existing ones the authors should cite more than 16 papers.

-Regeneration of the motor system takes long time in most of the experiments resulting in the growth and gaining weight of the animals (in rats it is significant). How does the package handle this phenomenon? If it is not able to do so yet, how will this correlation introduced?

-Both step length and step duration vary in healthy animals within wide ranges affecting the “style” of the gait. How does the dataset manage to handle these parameters?

Minor points

-Video-based semi-automatic gait analysing systems (e.g. DeepLabCut ) should be mentioned in the Introduction (with references) and they should be compared to the present method in the Discussion.

-Is there any possibility in the future to adopt this method to other systems different from CatWalk?

**Do you want your identity to be public for this peer review?** For information about this choice, including consent withdrawal, please see our Privacy Policy

Reviewer #1: No

---

## [Author Response · Author response to Decision Letter 1]

2 May 2025

Please find attached the revised version of our manuscript entitled “Runway: A Novel, Forward-Screen Method of Analyzing Catwalk Gait Analysis Data". We greatly appreciate the feedback offered on our previous submission and have taken steps to address any critiques or concerns raised.

In summary, we are greatly appreciative of the feedback supplied by both reviewers, and have incorporated their critiques into this revised manuscript. Please let us know if there remain any errors or unaddressed issues in this revised copy, and we will make the changes as soon as possible.

Below is a summary of the review feedback provided, annotated with our response.

The authors should support their statements with references in the Introduction much more often (e.g.: line 37, 44, 64). To find the place of this method among the existing ones the authors should cite more than 16 papers.

We agree that more citations are necessary in our instruction, and have added additional ones to further support our statements.

Regeneration of the motor system takes long time in most of the experiments resulting in the growth and gaining weight of the animals (in rats it is significant). How does the package handle this phenomenon? If it is not able to do so yet, how will this correlation introduced?

We agree that this is a significant factor in animal recovery post injury, and have added additional language starting at line 296 to address how our method can incorporate these variables.

Both step length and step duration vary in healthy animals within wide ranges affecting the “style” of the gait. How does the dataset manage to handle these parameters?

We agree that these factors are critical components, and have addressed them along with further discussing the importance of considering baseline performance with edits starting at line 166

Video-based semi-automatic gait analysing systems (e.g. DeepLabCut ) should be mentioned in the Introduction (with references) and they should be compared to the present method in the Discussion.

We strongly agree, and have included multiple reference to DeepLabCut and similar methods throughout the introduction and discussion sections.

Is there any possibility in the future to adopt this method to other systems different from CatWalk?

Yes, this is highly possible, and we have added edits starting at lines 73 and 296 to address this.

---

## [Decision Letter · Decision Letter 1]

20 May 2025

Runway: A Novel, Forward-Screen Method of Analyzing Catwalk Gait Analysis Data

PONE-D-25-04329R1

Dear Dr. Anderson,

We’re pleased to inform you that your manuscript has been judged scientifically suitable for publication and will be formally accepted for publication once it meets all outstanding technical requirements.

Kind regards,

Antal Nógrádi, M.D., Ph.D., D.Sc.

Academic Editor

PLOS ONE

Additional Editor Comments (optional):

Reviewers' comments:

Reviewer's Responses to Questions

**Comments to the Author**

Reviewer #1: All comments have been addressed

2. Is the manuscript technically sound, and do the data support the conclusions?

Reviewer #1: Yes

3. Has the statistical analysis been performed appropriately and rigorously?

Reviewer #1: Yes

4. Have the authors made all data underlying the findings in their manuscript fully available?

Reviewer #1: Yes

5. Is the manuscript presented in an intelligible fashion and written in standard English?

Reviewer #1: Yes

Reviewer #1: All my questions and comments have been addressed by the authors. The manuscript is eligible for being published in its present form.

**Do you want your identity to be public for this peer review?** For information about this choice, including consent withdrawal, please see our Privacy Policy

Reviewer #1: No
